# Stem Cell Therapy against Ischemic Heart Disease

**DOI:** 10.3390/ijms25073778

**Published:** 2024-03-28

**Authors:** I-Ting Tsai, Cheuk-Kwan Sun

**Affiliations:** 1Department of Emergency Medicine, E-Da Hospital, I-Shou University, Kaohsiung City 82445, Taiwan; tsai.iting@gmail.com; 2School of Medicine, College of Medicine, I-Shou University, Kaohsiung City 82445, Taiwan; 3Department of Emergency Medicine, E-Da Dachang Hospital, I-Shou University, Kaohsiung City 80794, Taiwan

**Keywords:** coronary artery syndrome, myocardial infarction, ischemia-reperfusion injury, ischemic cardiomyopathy, cellular cardiomyoplasty

## Abstract

Ischemic heart disease, which is one of the top killers worldwide, encompasses a series of heart problems stemming from a compromised coronary blood supply to the myocardium. The severity of the disease ranges from an unstable manifestation of ischemic symptoms, such as unstable angina, to myocardial death, that is, the immediate life-threatening condition of myocardial infarction. Even though patients may survive myocardial infarction, the resulting ischemia-reperfusion injury triggers a cascade of inflammatory reactions and oxidative stress that poses a significant threat to myocardial function following successful revascularization. Moreover, despite evidence suggesting the presence of cardiac stem cells, the fact that cardiomyocytes are terminally differentiated and cannot significantly regenerate after injury accounts for the subsequent progression to ischemic cardiomyopathy and ischemic heart failure, despite the current advancements in cardiac medicine. In the last two decades, researchers have realized the possibility of utilizing stem cell plasticity for therapeutic purposes. Indeed, stem cells of different origin, such as bone-marrow- and adipose-derived mesenchymal stem cells, circulation-derived progenitor cells, and induced pluripotent stem cells, have all been shown to play therapeutic roles in ischemic heart disease. In addition, the discovery of stem-cell-associated paracrine effects has triggered intense investigations into the actions of exosomes. Notwithstanding the seemingly promising outcomes from both experimental and clinical studies regarding the therapeutic use of stem cells against ischemic heart disease, positive results from fraud or false data interpretation need to be taken into consideration. The current review is aimed at overviewing the therapeutic application of stem cells in different categories of ischemic heart disease, including relevant experimental and clinical outcomes, as well as the proposed mechanisms underpinning such observations.

## 1. Introduction

Cardiovascular disease is the leading cause of mortality worldwide, with nearly half of these fatalities attributable to ischemic heart disease [1], which accounts for 12% of the loss of disability-adjusted life years annually [2]. The term ischemic heart disease encompasses a broad range of cardiac pathological conditions that represent different stages of the process of coronary artery obstruction, including acute and chronic coronary syndromes [1,3], myocardial infarction [4], ischemic cardiomyopathy [5], and ischemic heart failure from post-infarct myocardial remodeling [6], despite the lack of distinct boundaries between different categories (Figure 1). Among them, acute myocardial infarction, which is the first manifestation of ischemic heart disease in approximately 50% of patients [7], is a life-threatening situation that requires immediate treatment [e.g., percutaneous coronary intervention (PCI)] [4]. Current therapeutic strategies for less urgent ischemic heart disease include endovascular (e.g., percutaneous angioplasty and stenting) or surgical procedures (e.g., coronary artery bypass grafting) to maintain coronary arterial patency [8], as well as long-term pharmacological control [9]. Nevertheless, notwithstanding a 4.6% reduction in the global age-standardized prevalence rate in the last decade [10], the rates of mortality and morbidity from either acute attacks or the debilitating sequelae remain high, in spite of the latest refinement of preventive and therapeutic measures. The main reason for the progression of ischemic heart disease and its complications is the fact that cardiomyocytes undergo terminal differentiation and are irreversibly withdrawn from the cell cycle [11], despite findings of an increased mitotic index in the normal myocardial of an infracted heart [12] and a surge in circulating endothelial progenitor cells (EPCs) after myocardial infarction [13], suggestive of the possibilities of endogenous cardiomyocyte proliferation [14] and stem cell migration from the bone marrow [15]. Nevertheless, a study later demonstrated that these EPCs may originate from a niche in the vessel wall rather than from the bone marrow [16]. Although such endogenous restorative mechanisms, as well as current medical and mechanical therapeutic approaches, may decelerate disease progression, they cannot reverse its pathological changes [17,18]. Therefore, the idea that stem cells with regenerative potential may be utilized for treating cardiac diseases such as myocardial infarction and heart failure has sparked great interest in their potential clinical applications [19,20,21].

The proposal of cell transplantation for treating cardiac diseases, an approach known as “cellular cardiomyoplasty”, started two decades ago when the possibility of using multipotent adult bone marrow hematopoeitic stem cells and mesenchymal stem cells (or “mesenchymal stromal cells”, MSCs) for therapeutic purposes began to gain momentum [19,22,23]. The role of endothelial progenitor cells in adult neovascularization also started to attract academic attention [24]. Since then, myriad experimental studies have started to investigate the therapeutic efficacy of stem cells against different models of cardiac disease, as well as the underlying mechanisms [25,26,27,28,29,30]. Such promising therapeutic potential has also triggered the development of stem-cell-related cardiovascular tissue engineering, as reflected by a tremendous increase in the number of related research articles in recent decades [31]. Consistently, clinical trials focusing on the therapeutic use of stem cells for ischemic heart disease have followed the same trend. Although the first registered trial on cell therapy for coronary heart disease in 2002 was terminated, many have been successfully completed [30,32].

The aim of the present review was to provide an overview of the current status of stem cell application in the treatment of ischemic heart disease, the mechanisms involved, and the directions of future developments.

## 2. Stem Cell Therapies against Different Entities of Ischemic Heart Disease

### 2.1. Acute and Chronic Coronary Syndromes

The etiology of coronary artery disease (CAD) is complex, involving genetic and environmental factors. The progression of CAD, which is a dynamic process resulting from atherosclerotic plaque accumulation and alterations in the coronary blood flow, can have a relatively long stable period before becoming unstable, typically due to atherothrombotic erosion or rupture [33]. Regarding its clinical presentation, coronary syndrome can be categorized as either acute or chronic [34]. Despite the broad definition of acute coronary syndrome, with severity ranging from unstable angina to myocardial infarction, this section focuses on unstable angina, in which there is no biochemical evidence of myocardial infarction, despite the clinical symptoms being suggestive of acute coronary syndrome [1]. On the other hand, chronic coronary syndrome, more widely known as “stable coronary artery disease”, is defined as a reversible demand/supply mismatch leading to myocardial ischemia. Despite a possible misnomer, “stable” refers to a lack of symptoms or a condition being controlled following medications or revascularization. A history of myocardial infarction or evidence of coronary plaque formation on imaging studies (e.g., conventional or computed tomography angiography) is suggestive of this diagnosis [35].

Current therapeutic strategies for acute and chronic coronary syndromes include coronary revascularization with percutaneous coronary intervention (PCI) [36] or coronary artery bypass grafting [8], as well as medical treatments comprising antiplatelet therapy with additional secondary prevention measures such as intensive lipid-lowering therapy, neurohormonal agents, and a modification of lifestyle [1,35,37]. Nevertheless, despite being the gold standard, PCI was unable to improve all-cause mortality, hospitalization for heart failure, or the restoration of left ventricular function compared to medical treatment in the Revascularization for Ischemic Ventricular Dysfunction trial, which was a large-scale clinical trial recruiting up to 700 patients across 40 centers in the UK [38]. A review of the overall evidence also showed PCI’s lack of treatment advantage over conservative approaches in terms of all-cause mortality or myocardial infarction, unlike acute coronary syndrome [36].

With the increasing realization of the close correlation between the degree of endothelial dysfunction and cardiovascular diseases such as acute coronary syndrome, researchers began to study EPCs after discovering their role in ischemia-induced neovascularization, as well as endothelial regeneration following myocardial ischemic injury [39]. In contrast to stable coronary heart disease, the number of circulating EPCs significantly increases in patients with unstable angina, thereby highlighting their possible participation in the restoration of cardiac function through homing and engraftment in an ischemic myocardium [40,41]. Accordingly, ex vivo technology for the expansion of hematopoietic stem cells was developed [42]. The first randomized, double-blind, placebo-controlled trial aimed at assessing the effect of the intracoronary infusion of circulating progenitor cells on coronary macro- and microvascular function, as well as left ventricular function, was conducted in 2005 on 26 patients diagnosed with chronic coronary total occlusion after recanalization. The results showed an increase in coronary flow reserve by up to 43%. The number of myocardial segments with hibernation in the target region significantly declined at three months in the treatment group, in contrast to the absence of notable changes in the control group. Magnetic resonance imaging demonstrated a 16% reduction in infarct size coupled with a significantly elevated left ventricular ejection fraction by 14% in those undergoing cell therapies, due to an enhanced wall motion in the target region compared to the controls [43]. Following that clinical investigation, there have been a number of human trials focusing on the efficacy of intramyocardial human CD34+ cells injection in relieving symptoms and improving outcomes among patients with chronic angina refractory to optimal medical treatment or those not suitable for revascularization. A meta-analysis of those trials supported the therapeutic use of CD34+ cell therapy for cardiac ischemia after showing its superiority over placebos in reducing the risk of all-cause mortality and angina frequency, together with a prolongation of exercise time without increasing the incidence of adverse events [44]. Another more recent clinical trial further identified previous smoking habit, female gender, lower grades of angina score, and diastolic dysfunction as independent promising predictors of CD34+ cell therapy in patients diagnosed with end-stage diffuse coronary artery disease [45].

Besides hematopoietic stem cells (e.g., EPCs), it became evident two decades ago that other non-cardiomyogenic cells could also improve heart function, thereby presenting unprecedented opportunities for restoring cardiac function in patients diagnosed with ischemic heart disease [21]. In this aspect, bone-marrow-derived stem cells received particular attention [46]. Adult bone-marrow-derived stem cells were found to be able to differentiate into various cell types such as cardiomyocytes and endothelial cells. Experimental studies have shown that such a distinct property of stem cells, known as “plasticity”, not only could replace damaged heart muscle (i.e., myogenesis), but may also enhance cardiac revascularization (i.e., angiogenesis and vasculogenesis) [20]. The application of stem cells in the treatment of acute and chronic coronary syndromes before developing into overt myocardial infarction appears to be particularly promising because of their non-acute nature that allows for ex vivo autologous stem cell expansion [47].

In 2009, a randomized, double-blind, placebo-controlled trial on 50 patients diagnosed with chronic myocardial ischemia ineligible for conventional revascularization demonstrated a 3% absolute increase in left ventricular ejection fraction at three months, as well as an improvement in angina score and quality of life, in those receiving intramyocardial injections of autologous bone-marrow-derived mononuclear cells compared to the placebo group [48]. Apart from bone-marrow-derived stem cells, hematopoietic progenitor cells have been reported to have a positive part to play in the chronic heart ischemia setting. A prospective, randomized, double-blinded phase I clinical trial recruiting 38 patients with severe diffuse coronary artery disease refractory to medication and unsuitable for coronary intervention, who were subjected to intra-coronary CD34+ cell infusion after subcutaneous granulocyte colony-stimulating factor (G-CSF) administration twice a day for four days, showed highly significant improvements in left ventricular ejection fraction and increases in neovascularization, as well as the alleviation of angina and heart failure symptoms [49]. A subsequent phase II randomized controlled trial by the same group consistently demonstrated gradual and effective improvements in left ventricular systolic function one year after autologous CD34+ cell therapy in patients with diffuse coronary artery disease who were not candidates for coronary intervention [50].

A systemic review and meta-analysis in 2015, which included 48 randomized controlled trials recruiting 2602 patients diagnosed with either chronic ischemic heart disease or myocardial infarction, demonstrated an improvement in left ventricular ejection fraction and the suppression of myocardial remodeling in those undergoing adult bone marrow cell therapies [51]. In addition, a recent meta-analysis of 20 randomized controlled trials on the outcome and safety of intramyocardial stem cell transplantation during coronary artery bypass surgery reported a significant improvement in left ventricular ejection fraction and wall motion without increasing risk in patients receiving cell therapy compared to controls [52], thereby further supporting the significant therapeutic benefits of stem cell application in the treatment of ischemic heart disease.

### 2.2. Myocardial Infarction

Myocardial infarction is at the other end of the spectrum of coronary artery syndrome, representing a severe life-threatening condition that involves myocardial death. The well-known initiating mechanism underlying acute myocardial infarction is atherosclerotic coronary plaque erosion or rupture, causing exposure of the highly thrombogenic matrix in the plaque to the circulating blood [4]. Once infarction occurs, patients are at risk of complications, including infarct expansion, reinfarction, heart failure, a need for repeating revascularization, and mortality [53], given the non-regenerative nature of cardiomyocytes [11].

Based on the presence or absence of ST-segment elevation on electrocardiogram, acute myocardial infarction is categorized into ST-segment elevation myocardial infarction (STEMI) and non-STEMI. While STEMI results from a complete coronary artery occlusion and comprises approximately 30% of acute coronary syndrome, non-STEMI, which is caused by partial occlusion of the coronary artery or the presence of collateral circulation despite occlusion, accounts for approximately 70% of acute coronary syndrome [54]. According to cause, myocardial infarction is classified into six types, namely, infarction caused by coronary atherothrombosis (type 1), infarction attributable to a supply–demand mismatch not resulting from acute atherothrombosis (type 2), infarction leading to sudden death without biomarker or electrocardiographic confirmation (type 3), infarction related to PCI (type 4a) or coronary stent thrombosis (type 4b), and infarction linked to bypass grafting (type 5) [55].

Paradoxically, although the risk factor burden has been found to be positively associated with the income of a country [56], over 80% of mortalities from cardiovascular disease worldwide occur in low- and middle-income countries [57]. It is speculated that a higher economic status may help in the implementation of preventive and advanced treatment (e.g., revascularization) measures that contribute to an alleviation of the high burden of risk factors in higher-income countries [4]. Therefore, despite the emphasis on lifestyle modifications, these findings highlight the importance of effective preventive and therapeutic approaches. Current clinical guidelines recommend coronary catheterization and PCI within two hours of STEMI onset to decrease fatality from mortality from 9% to 7%, while prompt invasive coronary angiography followed by PCI or surgical revascularization within 24 to 48 h is suggested for those diagnosed with non-ST-segment elevation acute coronary syndrome to reduce mortality rate from 6.5% to 4.9% [54]. Despite the decreasing incidence of STEMI in high-income countries, as mentioned above [58], in-hospital mortality rates among patients with STEMI complicated by shock are still high, particularly in those with cardiac arrest [59]. In addition, the sequelae of myocardial infarction, including left ventricular dysfunction and remodeling, remain the major causes of morbidity among survivors [60].

Focusing on the therapeutic use of stem cells in this setting, experimental studies support the efficacy of injecting autologous bone-marrow-derived mononuclear cells into the area of myocardial infarction to preserve myocardium viability and left ventricular function following the induction of myocardial infarction in rat [61] and porcine [27] models. Consistently, the clinical potential of bone-marrow-derived mononuclear cells and MSCs has been explored [60]. Apart from bone-marrow-derived cells, experimental studies have revealed the effectiveness of adipose-derived MSCs in alleviating infarct size and preserving cardiac function after myocardial infarction [62,63,64,65].

On account of safety concerns about the allo- or xenotransfusion of stem cells for treatment purposes in clinical practice, the use of autologous stem cells remains the gold standard. On the other hand, unlike in experimental settings, the early administration of autologous stem cells after myocardial infarction in humans is impractical without the prior time-consuming process of ex vivo expansion [28,66]. To investigate the efficacy of a delayed stem cell treatment for myocardial infarction, fresh, uncultured, unmodified, autologous adipose-derived regenerative cells were delivered into a temporarily blocked coronary vein in an experimentally induced porcine chronic myocardial infarction model four weeks after infarction. The results showed a significant improvement in myocardial function, increase in myocardial mass, and reduction in scar tissue formation, indicating a therapeutic impact even at four weeks post-infarction [67]. The findings, therefore, suggested that the ex vivo expansion of autologous stem cells for delayed cell therapy may be a feasible clinical option for those experiencing myocardial infarction.

In a real-world scenario, a phase II, randomized, double-blind, placebo-controlled trial in the United States randomized 100 and 95 participants into a CD34+ cell therapy group and a placebo group, respectively, to investigate the safety and efficacy of autologous CD34+ cell intracoronary infusion in patients with left ventricular dysfunction (i.e., ejection fraction ≤48%) four days after successful stenting for STEMI [53]. Although increased perfusion was noted within each group up to six months after intervention, without significant differences in myocardial perfusion or adverse events between the two groups, a beneficial cell dose-dependent effect on left ventricular ejection fraction and infarct size, as well as a higher overall survival, were noted in subjects receiving mini bone marrow CD34+ cell harvest and administration after adjusting for the time of ischemia [53]. Hence, these findings support the safety and potential efficacy of bone-marrow-derived CD34+ cells in the treatment of patients with left ventricular dysfunction after STEMI [53]. Further clinical evidence is needed to compare the efficacy of different stem cells, as well as to identify the optimal dosage, route, and timing of administration. Furthermore, the clinical use of allo- or xenogeneic stem cells, bioengineering techniques, and methods to trace these implanted stem cells remain to be explored [68].

### 2.3. Ischemia-Reperfusion (IR) Injury

IR injury unavoidably happens when an organ is reperfused after being deprived of its blood supply, as in the case of revascularization (e.g., PCI, stenting, or bypass surgery) following a blockage of the coronary artery (i.e., myocardial infarction) [69]. Indeed, IR injury can account for up to 50% of the final infarct size after an episode of myocardial infarction [70]. Reperfusion of an ischemic organ triggers a cascade of inflammatory reactions, including the activation of both the innate and adaptive immune systems. While the former is characterized by complementary activation [71], the latter involves the recruitment of neutrophils and the subsequent generation of oxidative stress [72,73]. The resulting apoptosis, intracellular calcium ion overload, and alteration in cardiomyocyte energy metabolism contribute to cardiac function impairment [74]. Notwithstanding such an in-depth understanding of the mechanisms of IR injury, prior treatment attempts (e.g., supplementation of exogenous antioxidants) were found to be ineffective [75]. Therefore, survivors of myocardial infarction remain at risk of potentially fatal sequelae such as heart failure [76].

There is currently neither an effective prophylactic nor therapeutic strategy against IR injury of the heart, despite extensive research into identifying potential remedies and optimal treatment approaches [69,77,78,79,80,81,82]. On the other hand, the anti-inflammatory and anti-apoptotic effects of MSCs against organ IR injury in animal models were realized a decade ago [83]. A subsequent experimental study also demonstrated the therapeutic potential of adipose MSC-derived exosomes against cardiac IR injury [84], with the modulation of autophagy being reported as a potential mechanism [85]. Indeed, MSC exosome-derived noncoding RNAs have been proposed as a potential therapeutic tool [74]. Another animal study showed the cardiac functional benefit of enhancing the cell survival and VEGF expression of human amniotic membrane MSCs through the application of a biopolymer based on Poly(glycerol-sebacate)-co-Poly(caprolactone) (PGS-co-PCL) film on a rat model of myocardial IR [85], further underscoring the therapeutic potential of MSCs in this setting.

To date, a number of methods and agents have undergone clinical trials to test their efficacy against cardiac IR injury, such as remote ischemic pre- and postconditioning, dexmedetomidine, thymoglobulin, melatonin, and cyclosporine [32]. On the other hand, the results of previous clinical trials on the efficacy of stem cell application in alleviating cardiac IR are inconclusive and inconsistent [70], with just a couple of registered clinical trials currently focusing on the use of EPCs or MSCs; while the former is completed, the latter is still being prepared [32]. Therefore, clinical evidence supporting the use of stem cells in this setting is still pending.

### 2.4. Ischemic Cardiomyopathy

Ischemic cardiomyopathy refers to a compromised cardiac function characterized by a left ventricular ejection fraction of 35% or below in patients who have coronary artery disease [5]. Medical therapy with or without percutaneous coronary intervention (PCI)/coronary artery bypass graft (CABG) remains the mainstay of treatment [5,38]. However, a large-scale clinical study demonstrated no significant influence of PCI on subsequent arrhythmia-related mortality [38]. On the other hand, another large-scale clinical investigation showed an association between bypass surgery plus medical treatment and a lower incidence of death compared to those who received medications alone [5]. That study also demonstrated a positive correlation between the presence of a viable myocardium and an improvement in left ventricular ejection fraction, regardless of the choice of treatment strategy [5]. Paradoxically, such an improved myocardial viability and function had no significant impact on patient survival [5]. Taking into account that the therapeutic impact of stem cells is mainly on the improvement in cardiac contractility, as reported in a large-scale meta-analysis of 66 studies on animal models [86], the results of that clinical study [5] have raised questions regarding the actual clinical benefits of stem cell therapy, especially patient survival, in those with severely compromised cardiac function due to ischemic cardiomypathy.

Despite the absence of large-scale studies focusing on the efficacy of stem cell treatment against ischemic cardiomyopathy, a recent human clinical trial on three patients with severely compromised cardiac function tested the safety and feasibility of applying an allogeneic-induced pluripotent stem cell (iPSC)-derived cardiomyocyte patch in the treatment of ischemic cardiomyopathy [87]. That study showed an alleviation of heart failure symptoms without accompanying allograft-related adverse events in a follow-up of up to one-year. Additionally, the authors reported improved left ventricular contractility and myocardial blood flow three months after immunosuppressant administration, thereby providing a glimmer of hope for those for patients with severe ischemic cardiomyopathy refractory to conventional treatments [87].

### 2.5. Ischemic Heart Failure

Following myocardial infarction, a cascade of complex pathophysiological processes contribute to ventricular remodeling, resulting from not only alterations in ventricular size and shape, but also an impairment of cardiac function in response to various factors, including neurohormonal, mechanical, and inflammatory mediators, as well as those related to IR injury, energy metabolism, and genetic predisposition [76]. The subsequent development of ischemic heart failure, which accounts for 50% of all cases of heart failure [6], is associated with frequent and still increasing hospitalizations, as well as a five-year survival rate of less than 50% [88,89].

In this aspect, the efficacies of early revascularization (e.g., PCI and CABG) and medical strategies including fluid restriction and reinforcement of cardiac contractility, as well as medications (e.g., beta-blockers, diuretics, and angiotensin-converting enzyme inhibitors) and electrophysiological devices (e.g., cardioverter-defibrillators), in arresting disease progression remain limited [76,90]. Even gene therapy with gene-editing techniques (e.g., CRISPR-Cas9) has been proposed for correcting genetic mutations that contribute to the condition, despite its questionable benefits [90].

The idea of cell therapy for ischemic heart failure started two decades ago, speculating on the possibility of stimulating the proliferation of endogenous mature cardiomyocytes or resident cardiac stem cells, or the implantation of exogenous donor-derived or allogeneic cells (e.g., fetal/embryonic cardiomyocyte precursors, bone-marrow-derived MSCs, or skeletal myoblasts), as well as the introduction of angioblasts to regenerate the necessary capillary network [11]. In an attempt to repair or regenerate the ischemia-damaged myocardium, there have been different candidates, including feta/neonatal cardiomyocytes, embryonic stem cells, skeletal myoblasts, bone-marrow-derived stem cells, EPCs, peripheral-blood-derived CD34+ cells, cardiac progenitor cells, and fibroblasts, which apparently produced promising results in prior experimental studies on heart failure [91].

One of the earliest human clinical trials was published in 2003 to test the hypothesis that intramyocardial injections of autologous mononuclear bone marrow cells could improve cardiac function in 14 patients and 7 controls with severe heart failure. It demonstrated the safety of the procedure, as well as an improvement in ejection fraction and a reduction in end-systolic volume in the treatment group after a four-month follow-up [92]. On the other hand, a more recent multicenter, randomized, placebo-controlled trial on 19 patients diagnosed with chronic ischemic heart failure with limited stress-inducible myocardial ischemia demonstrated no significant differences in left ventricular ejection fraction and volume, myocardial perfusion, and functional and clinical parameters between the autologous intramyocardial bone marrow cell injection and placebo groups upon follow-up at 3 and 12 months, indicating no notable functional improvement in the cell treatment group [93]. Although the follow-up periods of both trials may have been too short to evaluate the impact of cell therapy on patient survival in the setting of ischemic heart failure, the unsatisfactory outcome of the latter trial may partly be explained by the finding of another recent study [94]. Given that bone marrow hematopoietic stem cells are believed to come to the rescue of the failing heart, a recent investigation comparing bone marrow composition and the circulating number of myeloid cells between individuals with heart failure and those without the condition revealed significantly fewer engrafted human CD34+ cells and decreased lymphoid cell production, as well as reduced survival, in a mouse myocardial infarction model after xeno-transplanting the bone marrow of those with heart failure to a null mouse model, despite an increased myeloid cell number compared with their non-heart failure counterparts. The results, therefore, suggest an adverse effect of heart failure on bone marrow composition and cell reconstitution potential, which may impair cellular responses to injury [94], thereby implicating a limited efficacy of using autologous stem cells for intramyocardial injections in patients with an advanced disease.

## 3. Origins of Stem Cells and Mechanisms Underlying Their Therapeutic Effects against Ischemic Heart Disease

### 3.1. Endogenous Stem Cell Differentiation

In contrast to the popular belief that the adult heart is a terminally differentiated organ without any regenerative capacity, evidence supports the presence of different populations of cardiac stem cells within the heart [95,96], which are characterized by the presence of c-kit surface receptors and considered to be able to differentiate into three major cardiac lineages, namely, myocytes, endothelial cells, and smooth muscle cells [97]. Indeed, prior studies have shown that the proliferation of existing cardiomyocytes, rather than the differentiation of progenitor cells, is the main source of cardiomyocyte renewal [98,99]. However, such cardiomyocyte proliferation is unable to produce more than 1% cardiomyocyte renewal per year in an adult human heart [99]. On the other hand, the finding of an increase in circulating EPCs following myocardial infarction [13] may imply endogenous cardiomyocyte differentiation as an alternate source of cell replenishment [14] (Figure 2). Nevertheless, a tangible clinical association of heart self-renewal with cardiac stem cells or EPCs has not been identified [99,100]. Therefore, despite the probable presence of a physiologic cardiac stem cell repair mechanism, the clinical significance of such a natural endogenous restorative capacity of the heart remains questionable [17,18].

### 3.2. Differentiation and Incorporation of Exogenous Stem Cells

In contrast to the physiologic recruitment of endogenous stem cells, myriad experimental and clinical investigations involving cellular transplantation have demonstrated the efficacy of administering autologous, allogeneic, and even xenogeneic stem cells in alleviating inflammation and preserving cardiac function in the setting of ischemic heart disease [101,102,103,104]. Given that cardiomyocytes derived from pluripotent stem cells have been shown to exhibit the typical characteristics of normal cardiomyocytes [105,106], the possibility of direct stem cell incorporation into the injured myocardium with subsequent differentiation into functional cardiomyocytes cannot be ruled out. However, unlike an animal model in which the non-invasive in vivo tracking of transplanted iPSC-derived cardiomyocytes was achievable up to six weeks [107], there is no reliable clinical evidence supporting myocardial stem cell incorporation in the setting of ischemic heart disease [101].

With growing evidence supporting the therapeutic impact of stem cell administration on ischemic heart disease, numerous studies have focused on the mechanisms underpinning the promising outcomes. Experiments using animal models of myocardial infarction revealed the suppression of inflammation (i.e., reduced IL-1beta/TNF-alpha/NF-kappaB expressions), oxidative stress (i.e., decreased NOX-1/NOX-2/oxidized protein expressions), myocardial hypertrophy (i.e., reduced β-myosin heavy chain expression), fibrosis (i.e., decreased transforming growth factor expression), and apoptosis (i.e., reduced cleaved caspase-3/cleaved PARP expressions), heart failure (i.e., decreased brain natriuretic peptide expression), mitochondrial (i.e., decreased cytosolic cytochrome C expression) and DNA damage (i.e., decreased gamma-H2AX expression), and an enhancement of angiogenesis (i.e., increased SDF-1alpha/VEGF expressions) after MSC treatment [102,108], highlighting the wide range of potential beneficial effects achievable through stem cell therapy.

Nevertheless, despite the acceptance of stem cell homing to the ischemia-injured heart as one of the mechanisms [109], the limited engraftment, retention, and survival of the transplanted stem cells raise a serious question about the actual role of direct stem cell participation in the repair process [110,111]. A previous meta-analysis of 43 randomized controlled trials investigating the efficacy of bone-marrow-derived cell therapy, mainly through intracoronary infusion against ischemic heart disease, showed that, despite a significant reduction in infarct size and increase in ejection fraction up to one year after treatment, no significant differences in infarct size and cardiac function were noted at three-year and five-year follow-ups [101]. Notwithstanding their finding of a superior five-year survival advantage of the cell therapy group compared to that of the control group, the results of that study suggested that the relatively short-lasting advantageous effects of cell therapy on infarct size and cardiac function may not explain the observed five-year survival benefit. In addition, none of the included trials demonstrated direct stem cell incorporation into the damaged myocardium [101], raising doubts about the contribution of direct cellular participation to the observed outcomes [99]. Furthermore, the finding that the in vivo introduction of successfully re-engineered human embryonic stem-cell-derived cardiomyocytes in non-human primate hearts would cause arrhythmia was not reported in previous clinical investigations, implying that the newly implanted stem cells in those trials may never have differentiated into functional cardiomyocytes [99].

### 3.3. Paracrine Effects and Exosomes

Taking into account ample evidence against the hypothesis that stem cell incorporation into the injured heart is a likely mechanism underlying the reported benefits of cell therapy for ischemic heart disease, the paracrine actions of stem cells began to receive attention [99]. Indeed, even without stem cell involvement, an early animal experiment demonstrated that the conditioned medium from culturing the normal cardiomyocytes of an infarct heart could preserve left ventricular function in a rat acute myocardial infarction model [26], highlighting the possible paracrine-related mechanisms from injured cells that contribute to the therapeutic benefits of cell therapy.

Extracellular vesicles (EVs), which are layers of lipid molecules released by somatic cells, participate in immune regulation, as well as tissue repair and proliferation [112,113]. Based on their size, EVs can be categorized into exosomes, microvesicles, and apoptotic bodies [114,115]. Exosomes, which are intraluminal vesicles 30–160 nm in diameter containing lipids, proteins, and nucleic acids, are generated from the endosomal system of various somatic cells [116,117,118] (Figure 3). Their precursors are endosomes, which, after being formed through membrane invagination (i.e., inward budding), later mature into multivesicular bodies (MVBs), before being released into the extracellular system to serve as mediators of intercellular communication, including intercellular signaling and metabolic functions via the delivery of their cargo to neighboring cells [119]. Exosome uptake by the recipient cells occurs through one of the three pathways: (1) interaction with surface receptors, (2) endocytosis, and (3) direct fusion with plasma membrane [120] (Figure 3). The release of exosomes is believed to be coordinated through the “endosomal sorting complexes required for transport” (ESCRT) system, which comprises four complexes of soluble protein (ESCRT-0, -I, -II, and -III) [121], as well as the associated accessory proteins (ALIX, VPS4, and VTA1) [122,123,124]. On the other hand, evidence suggests that they may also be generated via ESCRT-independent pathways [125].

Exosomes are different from other extracellular vesicles because of the presence of specific surface markers, such as Tsg101, CD9, CD63, and CD81 (i.e., the tetraspanin protein family) in the former [115,126,127], as well as the ability of exosomes to cross physiological barriers and their possession of distinctive immunologic properties [128]. Despite the possession of a tissue repair capacity similar to that of stem cells, exosomes have the merits of less tumorigenicity and immunogenicity, as reflected by their decreased probabilities of causing rejection and graft-versus-host reactions compared to stem cell therapy [129,130]. Serving as intercellular communicators [119], exosomes induce cellular phenotype differentiation and cytokine secretion [131]. In addition, exosomes have been shown to be therapeutic against ischemic heart disease through their ability to mitigate apoptosis [132,133], inflammation [134], and cardiac remodeling [135], as well as enhance angiogenesis [136]. The effects are compatible with those reported in previous experimental studies that used stem cells for treating myocardial infarction [102,137] (Figure 2).

One of the key components contributing to the therapeutic benefits of exosomes is microRNA (miRNA) [138], which is a single-stranded, short, non-coding RNA molecule consisting of 21 to 23 nucleotides [139]. Being ubiquitous in plants and animals, as well as certain viruses, miRNAs regulate post-transcriptional gene expression [140,141] by RNA silencing through cleaving mRNA strand into two parts, destabilizing mRNA by shortening its poly(A) tail, or translating mRNA into proteins [142,143]. Myriad miRNAs from stem-cell-derived exosomes have been identified as potential therapeutic agents against different entities of ischemic heart disease, ranging from atherosclerosis to heart failure [144] (Figure 3).

For instance, the MSC exosome-mediated delivery of si-LOC100129516 was found to enhance cholesterol efflux and reduce intracellular lipid accumulation, thereby suppressing atherosclerosis progression through the PPARγ/LXRα/ABCA1 signaling pathway [145]. In addition, stem-cell-derived miR-21a-5p [146], miR-125b-5p [147], miR-342-5p [148], and miR-145 [149] were reported to suppress atherosclerosis development. Once myocardial infarction occurs, stem cell exosome-derived lncRNA-UCA1 [150], miR-146a-5p [151], and miR-125b-5p [152] were shown to reduce myocardial apoptosis, while lncRNA KLF3-AS1 was found to decelerate the onset and development of myocardial infarction [153]. In addition, other exosome-derived miRNAs such as miR-125b-5p were demonstrated to reduce autophagy [154], while miR-183-5p and lncRNA H19 were found to improve cardiac function [155] and promote angiogenesis [156], respectively. Regarding IR injury, miRNAs from stem-cell-derived exosomes may also have a therapeutic role to play; while miR-22-3p was found to alleviate ischemic injury through suppressing the activity of the KDM6B-mediated BMP2/BMF axis [157], other stem cell exosome-derived miRNAs, including miR-221/222, miR-182-5p, and miR-143-3p, were shown to be protective against IR-induced myocardial apoptosis [158,159,160]. Even when heart failure has developed, studies have revealed that the exosome-derived let-7 family could improve heart function through suppressing fibrosis and adhesions [161], while miR-30e is both anti-apoptotic and anti-fibrotic [162]. Moreover, MSC exosome-derived miR-1246 was found to promote myocardial angiogenesis [163], which is a crucial protective mechanism against remodeling in ischemic heart disease [144] (Figure 3). Nevertheless, despite promising experimental outcomes, their clinical application in this setting remains to be elucidated. Likewise, small extracellular vesicles derived from iPSCs have been reported to have great potential in the treatment of cardiac injuries [164].

## 4. Directions of Future Research and Speculations

### 4.1. Pluripotent Stem Cells

Not only does tangible evidence support the feasibility of creating new cardiomyocytes through embryonic stem cells [165] or reprogramming of other cell types (e.g., cardiac fibroblasts) with cardiac transcription factors [166,167], but experimental studies have also shown that cardiomyocytes derived from pluripotent stem cells such as embryonic stem cells and iPSCs can form gap junctions, contract, and even express hormonal receptors essential for myocardial regulation [105,106]. Moreover, because individual-specific iPSCs can be acquired through the epigenetic engineering of somatic fibroblasts, the prospect of an unlimited supply of stem cells whose efficacy is unaffected by ageing is especially promising [168] (Figure 2). Regarding iPSC transplantation as a treatment for ischemic heart disease, the longer action potential and lower cell-to-cell coupling of human iPSC-derived cardiomyocytes than normal adult cardiomyocytes generate an electrophysiological gradient that predispose to arrhythmias following administration into recipient cardiac tissue [169], as previously reported [165]. Nevertheless, the technology of iPSC-derived cardiomyocyte has recently proven to be a powerful tool for understanding the genetic basis of myocardial electrophysiology, as well as correcting such a conduction anomaly [170]. In addition, it enables the development of in vitro models for exploring treatment strategies against other genetically predisposed arrhythmias [171]. Another study further showed that a vascular endothelial cell adhesion glycoprotein, cadherin-5, could enhance the differentiation of human iPSCs into sinoatrial node-like pacemaker cells, highlighting the possibility of developing biological pacemakers [172]. Focusing on the experimental results of iPSC-derived cardiomyocyte transplantation for ischemic heart disease, a recent meta-analysis investigating the efficacy and safety of this approach in animal models demonstrated significant improvement in left ventricular ejection fraction and reduction in left ventricle fibrosis area in animals with iPSC-derived cardiomyocyte treatment compared with those without [173]. More importantly, that study demonstrated no significant difference in mortality and arrhythmia risk between the two groups [173]. On the other hand, despite such a strong implication for clinical application, the transplantation of re-engineered cardiomyocytes (so called “second-generation stem cells” [30]) may accompany adverse complications such as tumorigenicity (e.g., teratoma) stemming from vector or cell pluripotency, which is another major concern when harnessing the plasticity of iPSCs in the treatment of ischemic heart disease [167]. Future investigations may focus on feasible means of striking a balance between the need for functional restoration and the risk of potential complications, as well as the long-term survival of the implanted cells in the cardiac environment.

### 4.2. Scaffold and Cell Sheet Approaches to Stem Cell Application

With advancements in tissue engineering in the recent decade, there has been a surge in research focusing on its application in the treatment of ischemic heart conditions [31]. A prior animal study showed that, instead of direct stem cell injection into the myocardium or through the coronary vasculature, the use of a patching approach with the heart infarct area covered by adipose-derived MSCs embedded in platelet-rich fibrin appeared to be a feasible alternative [137]. A recent experimental investigation also reported an improvement in post-IR cardiac function through the application of differentiated human amniotic membrane MSCs with a modified PGS-co-PCL film in a rat model, possibly attributable to enhancements in cell survival and VEGF expression [174]. Clinically, a recent human trial on three patients diagnosed with ischemic cardiomyopathy with severely impaired contractile function receiving an allogeneic iPSC-derived cardiomyocyte patch demonstrated improved ventricular contractility and myocardial perfusion [87], further highlighting the potential of a stem cell scaffold approach in the treatment of ischemic heart disease. With recent advances in artificial intelligence and three-dimensional printing, the development of an optimal scaffold for stem cell treatment may not be a far-fetched idea. On the other hand, the development of a scaffold-free MSC cell sheet through tissue engineering for the prevention of myocardial infarction and left ventricular remodeling seems to be another promising option [175].

### 4.3. Combination of Stem Cell with Other Strategies

Prior experimental and clinical studies have tried to explore the possible additional or synergistic effects achievable through combining stem cell treatment with other therapeutic modalities. For instance, although granulocyte colony-stimulating factor (G-CSF) is believed to be able to mobilize bone marrow stem cells to repair the ischemia-damaged myocardium through vasculogenesis [176], previous clinical trials failed to demonstrate any functional benefits of G-CSF administration both in patients with severe chronic ischemic heart disease [176] and in those with myocardial infarction after a relatively short follow-up period [177]. Another clinical trial with a follow-up of up to seven years also showed no survival benefit in patients receiving G-CSF after myocardial infarction [178]. However, a more recent clinical trial demonstrated a reduction in adverse left ventricular remodeling and an improvement in the quality of life in patients diagnosed with severe acute myocardial infarction having received G-CSF treatment after a 10-year follow-up [179]. Nevertheless, G-CSF was still part of the protocol in some clinical trials on patients with ischemic heart disease, taking into account its lack of overt adverse effects [180].

Apart from promoting the mobilization of endogenous stem cells, the long-term benefits of stem cell treatment against ischemic heart disease in the real-world clinical scenario depend on improvements in three key aspects, namely, the enhancement of stem cell homing, paracrine function, and survival [181,182]. For example, preclinical studies using animal models of myocardial infarction showed that not only could atorvastatin enhance the survival of the infused MSCs and improve cardiac function [183,184], but it could also facilitate the migration and homing of MSCs toward the infracted myocardium [185]. Furthermore, a number of experimental studies have investigated the benefits of hypoxic preconditioning and genetic modification, as well as other pharmacologic and non-pharmacologic pretreatments of stem cells to optimize their cardioprotection potential [181,182]. Regarding non-pharmacologic approaches, attention has been paid to the possibility of combining other therapeutic modalities that are known to promote angiogenesis and alleviate inflammation (e.g., shock wave [186]) with stem cell treatment against ischemic heart disease. Previous studies utilizing animal models of myocardial infarction have demonstrated that shock wave application, either after intramyocardial bone-marrow-derived MSC injection [29] or as a stem cell pretreatment strategy prior to cardiac implantation [187], significantly enhanced cardiac function compared to sham controls. Nevertheless, because such explorations are purely experimental due to safety concerns and the difficulty in acquiring data on long-term outcomes, their clinical significance remains to be elucidated.

### 4.4. Refinements of Paracrine and Exosome Treatments

With the lack of evidence supporting the direct myocardial incorporation and in vivo differentiation of stem cells after administration [99,101], the direction of stem-cell-mediated treatments against ischemic heart disease has turned to strategies aiming at reinforcing the efficacy of stem-cell-derived paracrine effects and exosomes (Figure 3). Certain unfavorable physical environments may be harnessed to promote exosome function. For example, considering that ischemia can be a trigger of cardiac repair mechanisms, the therapeutic effects of ischemic-preconditioned exosomes [188] and hypoxia-elicited MSC-derived exosomes [152] have been explored. Likewise, another animal study reported the pro-angiogenic properties of exosomes derived from hypoxia-inducible factor 1-alpha (HIF-1alpha)-modified MSCs [136].

Regarding pharmacologic enhancement, Tongxinluo, a traditional Chinese medicine for patients with angina [151], and atorvastatin [156] have been found to boost the therapeutic efficacy of MSC-derived exosomes in a rat model of acute myocardial infarction through noncoding RNA upregulation. Aside from pharmacologic manipulation, endogenous substances have also been reported to modulate exosome release. A study showed that adiponectin, a protein hormone secreted from the adipose tissue, could increase the biogenesis and release of exosomes from MSCs in a mouse model of heart failure [161]. Moreover, hemin, an iron-containing porphyrin and a potent heme oxygenase-1 (HO-1) inducer, was found to be cardioprotective against senescence through the action of miRNA from MSC-derived exosomes [155]. Besides hypoxia and pharmacologic and non-pharmacologic strategies, genetic modifications and several bioengineering approaches have been introduced to modify the miRNA contents and membrane protein components of exosomes in an attempt to maximize their therapeutic potential against ischemic heart disease [138] (Figure 3).

### 4.5. Allogeneic and Xenogeneic Stem Cells

To expand the source of therapeutic stem cells but avoid the tumorigenicity potential of iPSCs, the idea of using allo- and even xenogeneic stem cells in the treatment of ischemic heart conditions has sparked much clinical interest. Such a proposal has gained momentum after the finding of similar improvements in cardiac function between autologous and allogeneic stem cell treatment for ischemic heart disease in a meta-analysis of 82 animal studies [104]. Although the results seemed unsurprising, taking into consideration the findings of other studies that favored the paracrine effects of stem cells rather than their direct participation in the process [99,101], that study still shed light on the lack of immunologic complications in this setting. In regard to xenogeneic stem cell transplantation, an animal study showed that even early left intracoronary arterial xenotransfusion of human bone-marrow-derived MSCs into a Lee-Sung pig model of myocardial infarction could offer therapeutic benefits up to five months after infarction, as reflected by a reduction in infarct size and the preservation of LV function [102], further highlighting therapeutic mechanisms other than direct cellular incorporation into the infarct region, as well as the absence of adverse immunological complications. Clinically, a pilot human trial recruiting three patients with severely compromised ischemic cardiomyopathy who received iPS-derived cardiomyocyte patches with three-month immunosuppressive treatment demonstrated an improvement in the symptoms of heart failure without immunological complications in all three participants, as well as an enhancement of left ventricular contractility and myocardial perfusion in two out of the three patients, underscoring the safety and therapeutic potential of the approach [87]. Nevertheless, more evidence is needed before the validation of such a strategy as an acceptable clinical practice.

## 5. Conclusions

Except at the neonatal stage of development [189,190], the heart has a limited capacity for self-healing and functional restoration after ischemic injury, despite the limited endogenous repair mechanisms [98,99]. Despite the promising outcomes of stem cell treatment against ischemic heart disease in previous experimental and clinical studies, current evidence supports neither the incorporation of the infused stem cells into the injured myocardium, nor their in vivo differentiation into functional myocytes in the long run [99,101]. The observed stem-cell-mediated benefits have been shown to be attributable to the paracrine functions of stem cells. In this aspect, exosomes appear to play a key role, particularly because of their contents of microRNAs, which have been found to be of great clinical potential in the treatment of ischemic heart disease [118,144,191,192,193]. Taking into account the limited availability of autologous stem cells, exosomes from their allogeneic and xenogeneic counterparts may also offer significant therapeutic advantages [104]. Further investigations into possible genetic, pharmacologic, non-pharmacologic, and bioengineering modifications of exosome content to enhance their therapeutic potential are warranted to verify the clinical applicability of exosomes in this setting. In addition, the generation of novel cardiomyocytes from pluripotent stem cells (i.e., embryonic and induced pluripotent stem cells) may be another direction of exploration, despite the need for monitoring the potential adverse sequelae such as arrhythmia and tumorigenesis [165,167]. In summary, stem cell treatment against ischemic heart disease, a currently incurable entity, remains an uncharted territory in which sustained research efforts are still required to shed light on the optimal strategies after balancing the merits and risks in human trials.

## Figures and Tables

**Figure 1 ijms-25-03778-f001:**
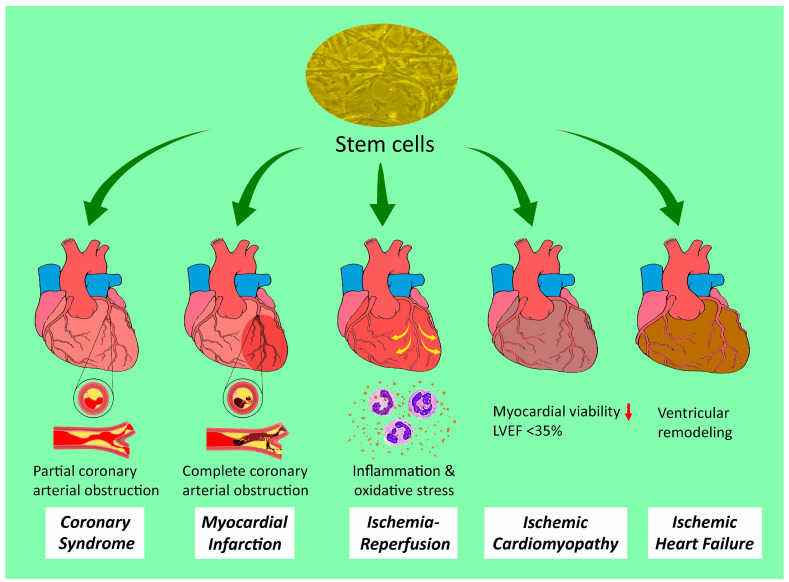
Therapeutic potential of stem cells against a spectrum of ischemic heart disease. LVEF: left ventricular ejection fraction. Red downwards arrow: reduction.

**Figure 2 ijms-25-03778-f002:**
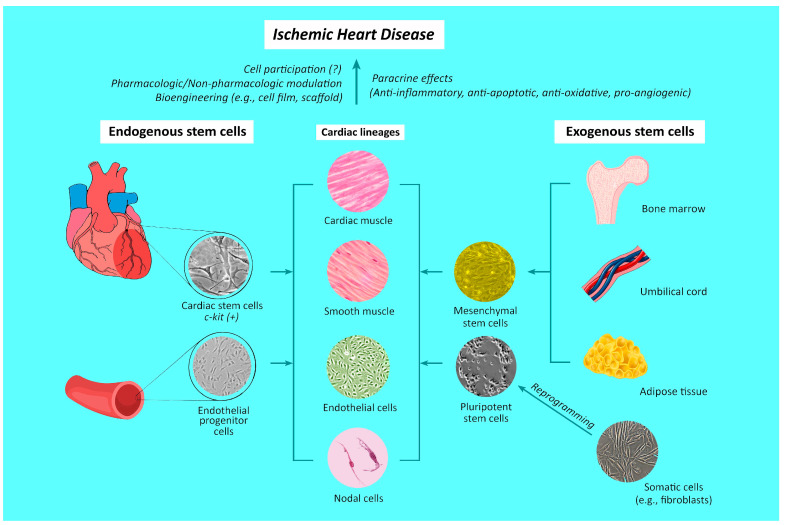
The reported origins of stem cells in the treatment of ischemic heart disease. Upwards arrow: therapeutic potential against.

**Figure 3 ijms-25-03778-f003:**
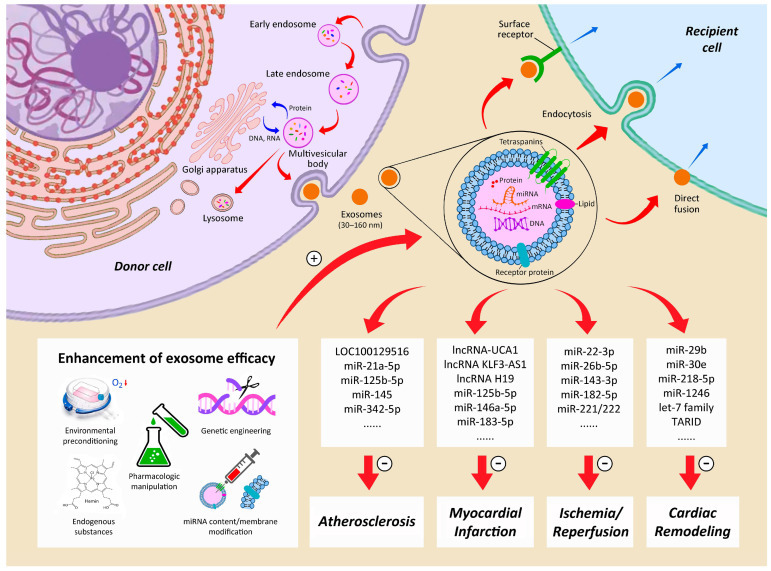
Stem-cell-derived exosomes and their therapeutic potentials against ischemic heart disease. miRNA: microRNA; positive sign: reinforcement; negative sign: suppression; small red downwards arrow beside molecular formula of oxygen: hypoxic environment.

## Data Availability

The original contributions presented in the study are included in the article, further inquiries can be directed to the corresponding author.

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
