# Peer review of "Stem Cell Therapy against Ischemic Heart Disease"

_ijms, 2024, doi:10.3390/ijms25073778_

Round 1

Reviewer 1 Report

Comments and Suggestions for Authors

This is a really well written review reports to highlight the development of stem cell therapies in the field of cardiac diseases. It is interesting to read and I found some aspects that I was not aware of. If I should suggest improvements, I would recommend to expand on recent finding with iPSC derived CMs for transplantation. There are novel studies and approaches to overcome arrhythmia induction. A further point in my mind: the application of hematopoetic cells from the bone marrow and MSC based therapies could be discussed more controversally in this review. It came out during the years that some positive reports result from fraud or false data interpretation. This makes it difficult to really understand which positive effects are there. 

Reviewer 2 Report

Comments and Suggestions for Authors

The manuscript reviewed various stem cell therapies from the view of heart medical doctors for treating ischemic heart diseases, which is very interesting and well described. However, it does not include very recent findings. Some of the sentences are difficult to understand because of the complicated description. The authors should write in a simpler way to be understandable.

 Lines 21-22, In recent two decades, the possibility that stem cell plasticity may be harnessed for therapeutic purposes has been increasingly realized. Clarify the sentence.

Lines 25-27, Besides, the paracrine effects accompanying stem cell administration has sparked a research interest in the underlying molecular mechanisms as reflected by recent intense investigations into the actions of exosomes. Clarify the sentence.

Lines 49-52, Nevertheless, notwithstanding the latest refinement of preventive and therapeutic measures, rates of mortality and morbidity from either acute attacks or the debilitating sequelae remain high despite a 4.6% reduction in the global age-standardized prevalence rate in the recent decade [10]. Clarify the sentence.

Line 127, what do “dormant segments” mean?

Lines 230-232, On the other hand, ex vivo expansion of stem cell population is a time-consuming process that precludes the possibility of early administration after myocardial infarction as in experimental studies [28, 66].  Clarify the sentence.

Line 325, what does “compromise of cardiac function” mean?

Figure 3, What symbolize and ?

Line 540, what is “severely compromised contractile function”?

Round 2

Reviewer 2 Report

Comments and Suggestions for Authors

I have no further comments.